# The Effects of High-Speed Resistance Training on Health Outcomes in Independent Older Adults: A Systematic Review and Meta-Analysis

**DOI:** 10.3390/ijerph19095390

**Published:** 2022-04-28

**Authors:** Alexandre Duarte Martins, Orlando Fernandes, Ana Pereira, Rafael Oliveira, Franco David Alderete Goñi, Nilton João Chantre Leite, João Paulo Brito

**Affiliations:** 1Comprehensive Health Research Centre (CHRC), Departamento de Desporto e Saúde, Escola de Saúde e Desenvolvimento Humano, Universidade de Évora, Largo dos Colegiais, 7000-727 Évora, Portugal; orlandoj@uevora.pt (O.F.); niltonchleite@gmail.com (N.J.C.L.); 2Sports Science School of Rio Maior, Polytechnic Institute of Santarém, 2040-413 Rio Maior, Portugal; rafaeloliveira@esdrm.ipsantarem.pt (R.O.); jbrito@esdrm.ipsantarem.pt (J.P.B.); 3Life Quality Research Centre, 2040-413 Rio Maior, Portugal; ana.fatima.pereira@ese.ips.pt; 4School of Education, Polytechnic Institute of Setúbal, 2910-761 Setúbal, Portugal; 5Research Centre in Sport Sciences, Health Sciences and Human Development, 5001-801 Vila Real, Portugal; 6Department of Physical Education and Sports, University of Granada, 18010 Granada, Spain; francod.agoni@gmail.com

**Keywords:** older people, high-speed resistance training, power training, neuromuscular function, health measures

## Abstract

Human ageing involves several physiological impairments—in particular, a decrease in sensorimotor function and changes in the nervous system reduce muscle strength, power, balance, and functional capacity performance. Preventive strategies are essential to ensure the quality of life of the elderly. High-speed resistance training (HSRT) may be an effective approach to muscle power development in this population, with significant short-term effects on neural adaptations and muscle power production. Therefore, the present study intends to analyze and systematize the studies focused on HSRT interventions and their effects on health outcomes in independent older adults. Four electronic databases (PubMed, Web of Science, EBSCO, and Scielo) were used for the purposes of searching randomized controlled trials that measured at least one key outcome measure focusing on velocity-based training and health outcomes in older adults on 7 March 2022 and identified 1950 studies. At the end of the process, fourteen studies were included in this systematic review and ten studies were included in the quantitative analysis. The main results showed that HSRT interventions would improve health measures, mostly cognitive function (large effects, *p* = 0.001, SMD = 0.94), neuromuscular function (moderate effects, *p* = 0.003, SMD = 0.70), and physical function (moderate effects, *p* = 0.04, SMD = 0.55 and *p* = 0.009, SMD = −0.59). Additionally, the results suggested that interventions with ten weeks or more, performed three times a week, provide significant improvements in neuromuscular function. In this sense, HSRT is effective for improving overall health outcomes in older adults. Future studies should include proper follow-ups (e.g., minimum six months) to assess the durability of HSRT intervention effects on all health-related variables.

## 1. Introduction

In accordance with the latest data report from Eurostat, approximately 20% of the total population in Europe is over 65 years old, and by 2050, this figure will rise to around 30% [1]. Human ageing involves several physiological impairments [2,3]. In particular, the decrease in the sensorimotor function and changes in the nervous system [4] reduce muscle strength, power, balance, and functional capacity performance [5,6], which can increase the risk of falling [7]. Falls are the second leading cause of unintentional injury-related deaths in the world population, and are becoming a serious public health problem for the elderly [8]. The United Kingdom spends an estimated 2.3 billion dollars a year in the treatment of injuries caused by falls [9]. Preventive strategies are essential to ensure the quality of life of the elderly and reduce healthcare costs, which threaten the stability of the public health system.

Falling or the fear of falling may impair physical activity levels and set in motion a vicious cycle in which the risk of future falls increases due to the deterioration of functional ability caused by not actively participating in daily activities [10,11]. The reduction in physical activity levels can have negative physiological effects for an ageing population (e.g., increased chronic diseases and the risk of non-communicable diseases) [12].

Due to reduced levels of physical activity and ageing, loss of muscle strength is associated with a decrease in the performance of daily life activities and negatively affects health, functional autonomy, survival, and quality of life, further increasing the risk of falling [13,14]. In this regard, the loss of muscle strength may be more closely associated with the impairment of muscle power than muscle mass [15,16,17,18]. On the other hand, muscle power has been reported to decline faster with ageing when compared to muscle strength [17,19].

High-speed resistance training (HSRT) may be an effective approach to muscle power development in this population [20,21,22,23,24,25], with significant short-term effects on neural adaptations and muscle power production [15,25,26]. Several studies [24,27,28,29] suggest that the above-mentioned intervention has an impact on the power generation capacity of older adults. In this sense, the HSRT approach is a key intervention strategy to counteract the harmful effects caused by ageing [28,30,31,32,33]. Some studies [34,35] reported that this method also makes it possible to mitigate capillary deterioration resulting from ageing, especially in the lower limbs. A suitable capillarization can improve protein synthesis metabolism by ensuring the transport of amino acids and growth factors to muscle fibers—in particular, growth factor-1 (IGF-1), hepatocyte growth factor (HGF), interleukin-6 (IL-6), and myostatin—thereby regulating satellite cell function and supporting muscle repair and/or remodeling/adaptation, which are essential after 65 years of age [36,37]. In addition, this intervention can offer advantages concerning other traditional resistance training (RT) interventions, mainly when the participants perform the concentric muscle contraction of each exercise repetition as rapidly as possible, showing improvements in morphological and neural adaptations [25,38] and functional capacity performance [39].

To our knowledge, solely two systematic reviews studied the effects of interventions of HSRT on health outcomes in older people, such as functional capacity [40] and power outcomes [41]. However, said studies were conducted in participants aged over 60, compared the effects caused by fast-intended velocity vs. moderate and low-velocity resistance training, and did not compare the RT program’s effects on the experimental group (EG) vs. the control group (CG). Therefore, it is relevant to carry out a systematic review that summarizes the effects of HSRT interventions on a population over 65 years of age and compare the EG and the CG in the most diverse parameters related to health status, such as body composition parameters, phase angle, heart rate variability, balance, and gait variability. Another significant criterion used in this review is that it only includes independent participants (i.e., older people that are able to perform activities of daily living independently, notably eating, bathing, and mobility) [42,43].

Therefore, the present review intends to analyze and systematize the studies focused on HSRT interventions and their effects on health outcomes in independent older adults. In this regard, health outcomes are defined as events that occur with an end result after an intervention. These outcomes can be measured clinically through physical examinations, laboratory and imaging tests, self-reported through questionnaires, or observed, such as fluctuations in gait pattern [44]. Thus, the present review study and meta-analysis sought to systematize the effects of HSRT interventions on health outcomes. Finally, we hypothesized that the HSRT interventions would improve health status parameters and that longer-duration HSRT interventions provide more significant benefits than short-duration HSRT interventions.

## 2. Materials and Methods

This systematic review and meta-analysis was conducted in accordance with the items of the Preferred Reporting Items for Systematic Reviews and Meta-Analyses (PRISMA) [45]. This review was also protocoled at the International Prospective Register of Systematic Reviews (PROSPERO), registered under number CRD42021272242.

### 2.1. Search Strategy and Data Sources

The following electronic databases were used to search for relevant publications on 7 March 2022: PubMed, Web of Science (WoS), Psychology and Behavioral Sciences Collection (EBSCO), and SciELO Citation Index (Scielo CI). In order to include the most recently published studies in the review, we set citation notifications for all these databases, but no restriction to date of publication was applied.

The data search was performed by using the U.S. National Library of Medicine’s Medical Subject Headings terms and English language terms related to HSRT and the elderly population. The following word combinations were used in all databases: (“elderly” OR “older people” OR “older adult” OR “older subject”) AND (“high-speed resistance training” OR “high-speed strength training” OR “high-speed power training” OR “power-training” OR “velocity-based training” OR “rapid-strength” OR “explosive-strength”). It was possible to perform 28 different searches in each of the mentioned databases (112 in total) through these word combinations. Additionally, the reference lists of the studies retrieved were manually searched to identify potentially eligible studies not captured by the electronic searches (1 study was found).

### 2.2. Eligibility Criteria

The inclusion of studies for revision was decided by consensus between the first and last authors. In case of disagreement regarding the inclusion of an article, the second author was consulted. The selected studies were then filtered according to the following inclusion criteria: (1) studies included participants with age ≥ 65 years old, from both sexes, without any clinical condition and physically independent; (2) at least one of the protocols under study was an HSRT program with maximal concentric velocity (i.e., as fast as possible, ≤1 s) lasting 2 weeks or longer; (3) studies included pre- and post-intervention measurements; (4) the study compared the EG with the CG at post-intervention; and (5) only original and full-text studies written in English, Portuguese, or Spanish.

Trials were excluded if: (1) studies included participants with age < 65 years old; (2) participants performed an intervention other than HSRT; (3) did not compare the results of the EG with the CG; (4) did not clearly describe the concentric and/or eccentric velocity; (5) examined of the effects of combined training methods (i.e., fast-intended- and moderate-velocity or balance training); (6) included participants with any associated disease (i.e., cancer, dementia, diabetes) or non-physically independent; (7) written in a language other than English, Portuguese, or Spanish; and (8) other article types than original (e.g., reviews, letters to editors, trial registrations, proposals for protocols, editorials, book chapters, and conference abstracts).

### 2.3. Data Extraction

All articles identified by the search strategy underwent an evaluation of the titles and abstracts, in duplicate, by two assigned researchers. All studies that did not meet the inclusion criteria were excluded. Abstracts that did not provide sufficient information regarding the inclusion and exclusion criteria were selected for full-text evaluation. In a second phase, the same two assigned researchers independently evaluated all selected full-text articles and conducted a second selection following the inclusion and exclusion criteria. In a meeting with a third researcher present, disagreements between reviewers were solved by consensus.

Data extraction from the selected full-text articles was conducted by both assigned researchers independently, using a list of intended data: (i) first author; (ii) year of publication; (iii) the sample (size and age); (iv) intervention details—duration, frequency, intensity, volume (sets and repetitions), rest interval, concentric and eccentric velocity, and type of exercises; (v) control group characteristics; and (vi) main results (health status).

### 2.4. Methodological Quality Assessment

The methodological study quality was assessed using the Physiotherapy Evidence Database (PEDro) scale [46], a valid and reliable instrument to assess eligibility, group allocation, blinding of allocation, and comparison between groups at baseline and its outcomes. This scale includes 11 questions with yes or no answers (yes = 1; no = 0), providing a total score that ranges between 0 (poor methodological quality) and 10 (excellent methodological quality) (the first item is not included in the rating).

Scores were obtained from the PEDro database and were therefore scored independently, avoiding any potential bias of the authors. When a study was not available on the PEDro database, two authors alone (A.D.M. and R.O.) rated the risk of bias. Disagreements between authors were solved by consensus in a meeting with the last author present (J.P.B.).

### 2.5. Level of Evidence

Based on the physiotherapy evidence database scale and to assess the interventions’ evidence, the Van Tulder criteria [47] were applied. Therefore, the selected studies were grouped by levels of evidence, according to their methodological quality. A study with a physiotherapy evidence database score of 6 or more is considered level 1 (high methodological quality) (6–8: good, 9–10: excellent) and a score of 5 or less is considered level 2 (low methodological quality) (4–5: moderate; <4: poor).

Due to the clinical and statistical heterogeneity of the results, a qualitative review was performed, conducting a best-evidence synthesis [48,49]. This classification indicates that if the number of studies displaying the same level of evidence for the same outcome measure or equivalent is lower than 50% of the total number of studies found, no evidence can be concluded regarding any of the methods involved in the study.

### 2.6. Statistical Analysis

When the same variable was found in more than two articles, it was included in the meta-analysis. In this sense, the studies included in this analysis (n = 10) were analyzed using the random effects model [50]. The forest plot was generated for some health outcomes. Studies’ heterogeneity was assessed by calculating the following statistics: (i) Tau^2^, (ii) Chi^2^, and (iii) I^2^. The following classification was used to evaluate the I^2^ (i.e., described inconsistency between trials): lower than 50% represents low heterogeneity; 50–74% represents substantial heterogeneity; and 75% and higher represents considerable heterogeneity [51].

The following scale was used to evaluate the standardized mean differences (SMDs) (i.e., SMDs = ([mean post-value intervention group—mean post-value control group]/ pooled variance) [52], according to the following thresholds: 0–0.19 = negligible effect; 0.20–0.49 = small effect; 0.50–0.79 = moderate effect; and 0.80 and higher = large effect.

All statistical analyses were performed within the Cochrane Review Manager (RevMan) [Computer program], version 5.4.1, The Cochrane Collaboration, 2020 [53].

## 3. Results

### 3.1. Studies Included

A total of 1950 studies were retrieved from the selected databases and one study was found in other sources, using the chosen keywords. These studies were then exported to reference manager software (EndNoteTM 20.0.1, Clarivate Analytics, Philadelphia, PA, USA). Duplicates (1540 references) were subsequently removed either automatically or manually. The remaining 411 articles were screened for their relevance based on titles and abstracts, resulting in the removal of a further 209 studies. At the end of the screening procedure, we had 202 eligible articles, which were read and analyzed in depth. After reading full texts, a further 188 studies were excluded due to not meeting the eligibility criteria (Figure 1). A total of fourteen studies were identified as meeting the criteria for inclusion and were assessed for quality using the PEDro scale. Finally, ten of these articles were included in the quantitative analysis (meta-analysis).

### 3.2. Methodological Quality

The methodological assessment of the fourteen studies included can be found in Table 1. These studies obtained a score between 3 [54,55,56,57] and 6 [58,59] on the PEDro scale. From all studies reviewed, two studies presented high methodological quality (level 1) [58,59], and twelve studies showed low methodological quality (level 2) [54,55,56,57,60,61,62,63,64,65,66,67]. Furthermore, the results of the PEDro scale showed that five studies did not specify the inclusion and exclusion criteria [55,56,57,65,66], three studies did not randomly allocate the participants into groups [54,55,56], two studies did not perform a baseline comparability [57,60], and all studies allocated the participants into groups in a concealed way.

Additionally, none of the studies included blinded the participants and the technicians responsible for the program’s sessions, two studies blinded the technicians who measured at least one key outcome [58,61], seven studies could not perform assessments on at least one key outcome from more than 85% of subjects [57,61,62,64,65,66,67], only one study performed an intention-to-treat analysis [59], three studies did not report a between-group statistical comparison for at least one key outcome [54,55,56], and finally, all studies reported both point measures and measures of variability for at least one key outcome.

### 3.3. Studies’ Characteristics

Table 2 shows the characteristics of the studies included in this systematic review in terms of country, sample, age, body fat, body mass, and BMI. Most of the included studies did not register the intervention, and only one study reported the protocol [61]. Two studies [58,66] only included females in their analysis and no study was conducted with only men.

Regarding the origin of the studies, six studies were conducted in the United States (North of America) [58,59,60,62,64,65], three were performed in Germany (Europe) [54,55,56], three were conducted in South Korea (Asia) [63,66,67], and the last two articles were carried out in two different countries on the same mainland (Europe)—Denmark [61] and Greece [57]. Fourteen studies included in this review randomized 408 participants into two groups; thus, CG included 199 participants and EG (were performed HSRT) included 209 participants. These participants ranged in age from 69 years old [54] to 82 years old [61].

### 3.4. Interventions’ Characteristics

Table 3 presents the description of the interventions performed in the included studies, namely exercise modality, names of exercises, weekly frequency, intensity, sets, reps, duration of the intervention and sessions, eccentric velocity, and whether the sessions were supervised or not.

The duration of the RT interventions ranged from eight [60,63] to sixteen weeks [57,67], with two studies reporting for eight weeks [60,63], three reporting for ten weeks [54,55,56], seven reporting for twelve weeks [58,59,61,62,64,65,66], and two reporting for sixteen weeks [57,67]. Concerning the duration of sessions, these varied from 30 [58] to 60 min [64,66,67]; the intensity of the sessions mainly was quantified through the percentage of repetition maximum (RM) [57,59,60,61,62,64,65]. For the majority of the studies, the sessions were performed three times a week [54,55,56,57,58,59,60,62,63,64,65,67].

The research seems to be relatively diverse regarding the number of sets and repetitions. The number of sets performed in the interventions was as follows: ten interventions used three sets [54,55,56,57,58,59,61,62,64,65], three interventions used two to three sets [63,66,67], and only one intervention used a non-fixed number of sets [60]. Regarding the number of repetitions used per set of exercises, three studies used six to ten [54,55,56], three studies used eight to ten [57,62,64], two studies used ten to twelve [60,63], one study used twelve to fourteen [65], two studies used twelve to fourteen [66,67], one study only used eight [59], one study only used ten [58], and one study used a non-fixed number of repetitions [61].

The exercise interventions of the fourteen studies included in this systematic review were performed using elastic bands [63,66,67], in a weightlifting machine [54,55,56,61], using computerized pneumatic machines [57,59,60,62,64,65], and using a weighted vest [58].

Ten studies reported the dropout rates of their respective participants [54,56,57,58,61,62,63,64,65,66], with two studies stating that no participants dropped out of the EG [58,64]. Furthermore, seven studies presented the dropout rate of its EG with further explanations [54,56,57,61,63,65,66] and one study provided no explanation for said dropout [62]. Two studies did not report adverse events in both groups [58,63]. However, three studies reported several negative events in both groups, notably falls that did not result in fractures (three cases in EG and CG), and temporary localized joint pain (three cases in EG and one case in CG) [64]. One of the three studies mentioned above reported sacral-iliac pain following completion of baseline testing in CG, as well as persistent chest pain in EG [59], and another study reported the occurrence of serious medical events [66]. Additionally, some participants from the groups mentioned above withdrew from the studies due to the diagnosis of several types of tumors and aortic aneurism [59,64]. Finally, supervised exercise interventions were used in most studies (Table 3).

### 3.5. Results of the Studies

Table 4 presents the aim, outcome variable, and main results of the fourteen studies in this review. Only outcomes addressed in more than three studies will be described here.

Physical function was analyzed in seven studies [54,57,58,61,63,66,67]. Four studies showed improvements in physical function in only the EG between pre- and post-training [58,61,63,67]. Outcomes of muscle strength, such as grip strength, were assessed in six studies [56,57,61,64,66,67]; three out of seven studies [61,64,67] reported differences between CG and EG at post-training in these outcomes (*p* < 0.05). Lastly, outcomes of muscle power were described in four studies [54,55,59,65]; of these four studies, two [59,65] demonstrated significant differences between CG and EG at post-training in peak power outcomes at different %RM (40 to 90% 1 RM) (*p* < 0.05).

### 3.6. Meta-Analysis

The meta-analysis with ten studies showed moderate to large effects of the HSRT interventions on some health outcomes (cognitive function, neuromuscular function, and physical function) in older adults. Four studies revealed large effects on cognitive function (analyzed through frontal assessment battery (FAB) and mini-mental state examination (MMSE)) in favor of intervention groups compared to control groups (*p* = 0.001, SMD = 0.94 [0.20, 1.68], *I*^2^ = 71%); see Figure 2.

In the same sense, six articles indicated moderate effects on neuromuscular function (analyzed through grip strength and muscle and power outcomes) in favor of intervention groups compared to control groups (*p* = 0.003, SMD = 0.70 [0.24, 1.15], *I*^2^ = 53%); see Figure 3. Finally, the analysis of the effects on physical function was divided into approaches in order to facilitate the interpretation of the results (Figure 4A,B). Consequently, this figure shows moderate effects on physical function in favor of intervention groups compared to control groups (Figure 4A, *p* = 0.004, SMD = 0.55 [0.02, 1.07], *I*^2^ = 72%; and Figure 4B, *p* = 0.009, SMD = −0.59 [−1.27, 0.08], *I*^2^ = 35%).

## 4. Discussion

The present systematic review and meta-analysis aimed to analyze and systematize the studies focused on HSRT interventions and their effects on health outcomes in independent older adults. The main findings suggest a lack of good-quality and pre-registered studies comparing HSRT interventions and CG. These analyses revealed moderate to substantial levels of heterogeneity and a wide range of predictive interval. There were moderate to substantial levels of heterogeneity between the studies. High heterogeneity may indicate that there are no benefits of HSRT over control groups [51,69]. Furthermore, all the variables show a prediction interval crossing the zero line.

The main finding of this review suggests that HSRT interventions would improve health measures—mostly cognitive function (large effects), neuromuscular function (moderate effects), and physical function (moderate effects)—thereby confirming the study hypotheses.

It is also important to note that there is a relationship between the duration of the intervention and the level of benefit. In this regard, interventions with ten weeks or more revealed significant improvements in muscle strength and power [54,55,56,59,61,64,65,66,67] and physical function [58,61,63,66,67]. These findings are of extreme importance for clinical practice, since muscle power is essential for physical performance during daily tasks [70,71].

For the sake of clarity, the Discussion section is organized into subsections according to the variables studied.

### 4.1. Cognitive Function

Regarding cognitive function, only one study investigated the retinal microvasculature density through changes occurring in the brain in cognitively normal older people [60]. The authors reported a negative correlation between retinal microvascular density and cognitive function (r = −0.54; *p* = 0.007), indicating that the improvement in cognitive function after HSRT could be mediated by the vascular effects. Nonetheless, regarding cognitive function, the studies by Yoon et al. [66] and Yoon et al. [67] addressed the effects of HSRT on cognitive function through the Montreal cognitive assessment (MoCA), FAB, and MMSE tests. The authors showed positive effects on processing speed, cognitive assessment, FAB, and MMSE test parameters. However, Lee et al. [63], who reported positive effects of 8-week HSRT on the neuromuscular function and gait performance in the elderly with mild cognitive impairment, failed to find any increase in the executive function of the frontal lobe. The sample characteristics and time of intervention may have compromised the expected results for these outcomes. Other studies have reported positive effects of HSRT on processing speed and executive function [67,72]. However, current evidence is limited.

The small number of included studies reduced the statistical power of this meta-analysis and did not allow an appropriate analysis of heterogeneity. Hence, more research on the role of exercise parameters (e.g., volume, types, and intensity) in specific cognitive functions is necessary. Addressing cognitive function is particularly important when considering the decline in executive and functional ability associated with ageing [73,74,75,76]. In this sense, cognitive function and muscular power [6,77,78] should be considered the primary factors influencing the decline in functional status in elderly populations. Consequently, it is fundamental to find ways to attenuate age-related executive function and muscular power decline due to the significant impact on life quality and the maintenance of an independent lifestyle.

Since neuromuscular deterioration accompanying the process of ageing is one of the first signs of decreased power output [78,79] and atrophy in areas associated with neuromuscular control [75], a potential relationship may exist between age-related declines in muscular power and cognitive function. Remarkably, research shows that exercise programs enhance a variety of functional and cognitive tasks, and that brain regions exclusively dedicated to executive function appear to be especially sensitive to exercise training [80,81,82,83]. The HSRT interventions showed promising results in cognitive executive function and memory, cognitive flexibility, and working memory [63,84]. The intensity of these interventions can be controlled through changes in load and movement velocity. However, research that examined resistance training and cognition used loading to increase the intensity, which thus justifies further studies controlling the speed or using both load indicators [63].

### 4.2. Neuromuscular Function

Concerning neuromuscular function with a particular focus on neural functions, some studies investigated the effects of HSRT on neural factors [54,55,56,61,63]. The effects of the HSRT intervention on neural factors showed positive results in two aspects: first, between the CG and EG in movement time, motor time, rate of electromyography (EMG) rise, rate of torque development, and percentage of voluntary activation; and second, between pre- and post-intervention in normalized parameters peak torque, and in the EMG amplitudes of the knee flexors, extensors, vastus lateralis (VL), vastus medialis (VM), and rectus femoris [56].

In the latter study [56], muscle activation determined by recording the surface electromyogram during a maximal isometric leg exercise demonstrated that the rates of EMG rose, similar to other studies. In this sense, Reid et al. [85] reported a significant increase in the rate of EMG rise among the elderly averaged 77-year old who performed HSRT at 40% of 1 RM for 16 weeks compared to the elderly who underwent HSRT at 70% of 1 RM. It can be inferred that the low-intensity HSRT may influence the apparent increase in the rate of EMG rise. This increase may occur because the abilities to recruit motor units and maximal motor unit-firing rate were developed in parallel to the increased rate of torque development associated with recruiting motor units [86].

In the last two decades, studies with pneumatic and isokinetic devices have shown that the peak muscle power of the leg extensors [54,55,56,59,63,65,66,67,87,88] and ankle plantar flexors [18] are stronger predictors of functional performance than muscle strength. The peak muscle power declines sooner and more rapidly than muscle strength [19,89] and is a more important predictor of functional performance than strength [18,87]. Therefore, peak muscle power could be a more critical variable on which to focus RT programs among older men and women.

Based on data from selected studies [56,61,63], this review reveals that HSRT was more effective than traditional resistance training with rising muscle activation, as reported by Reid et al. [85]. There was an increase in muscle activation of VM and VL muscles, suggesting increased recruitment and firing frequency of motor units [55,86,90,91]. In addition to gains in maximal EMG amplitudes and isometric strength, older adults showed elevated knee extensor activation and coactivation during early stance and elevated plantar flexor activation during push-off [55]. The data suggest that power training-induced increases in voluntary muscle activation and agonist muscle activation underlie increases in isometric muscle strength and gait velocity. Therefore, training based on HSRT can modify neuromuscular activation and increase older adults’ leg muscle strength and walking performance.

The small number of included studies for grip strength reduced the statistical power of this meta-analysis and there were substantial levels of heterogeneity between studies. However, all the variables showed a prediction interval crossing the zero line, suggesting the possibility of the real beneficial effects of HSRT. For the four studies on the muscle and power outcomes, null heterogeneity may indicate that there are real benefits of HSRT over the CG.

The mechanism underlying the increased neuromuscular activation may involve, but is not limited to, increased neural drive via corticospinal pathways, increased motor neuron and/or muscle fiber excitability, an increased number of active motor units, and/or increased conduction velocity [92,93].

The neuromuscular function was evaluated through various strength parameters, such as grip strength and isokinetic peak torque/body weight [66,67], rate of torque development [63], isokinetic power in average at different execution speeds [67], isometric strength [56], peak power at 40 to 80% 1 RM [65], and knee extension and leg press strength [64].

Moreover, the maximal isokinetic power at different execution speeds was evaluated in relation to body weight and joint range [54], and one study evaluated maximal isometric strength relative to leg length [61]. Two studies evaluated peak power at different percentages of 1 RM [64,65], as well as the velocity at peak power in air resistance equipment [65]. The maximal strength was evaluated through the 1 RM in leg extension, leg press [59,64], and chest press exercises [57]. Our findings evidenced that HSRT interventions are effective in improving strength and power. Despite the consistency in the results of the assessments on neuromuscular function in the comparison of pre–post or/and between CG and EG, some studies do not report positive effects in some parameters [57,59,66]. In the study by Reid et al. [59], the 1 RM of knee extension increased significantly from baseline in either the strength training group (41%) or the HSRT group (49%) when compared to CG (*p* < 0.01). However, there was no significant time–group interaction for the execution of the 1 RM leg press exercise. Hence, the authors concluded that the principal finding of their study was that, in older people with mild–moderate mobility impairments, a 12-week HSRT intervention induced similar improvements in lower-extremity muscle power compared to traditional slow-velocity resistance training. The non-significant increases in absolute power on the leg press in response to HSRT are considerably lower than those reported by other studies in older adults [57,64,65,94]. These results suggest that gains in muscle quality, which occurred without measurable hypertrophy, may have been due to neural adaptation to this form of explosive resistance training [63,95]. The muscle power decreases more precipitously than muscle strength in older adults, causing a dramatic loss in the ability to produce force rapidly.

The current evidence emphasizes the challenges regarding the development of optimal exercise strategies for clinically relevant outcomes in older adults [96,97]. In this sense, strategies that promote rapid increases in strength development should be a top priority in resistance training interventions, followed by methods that stimulate muscle hypertrophy [98,99].

### 4.3. Physical Function

Regarding physical function, namely gait function, over the past two decades, there has been a proliferation of research surrounding resistance training to increase functional activities. Specifically, studies have demonstrated that HSRT can promote significant improvements in functional outcomes [57,100,101,102] compared to standard resistance training programs.

Nine of the studies in this systematic review used some form of a timed walking test, namely the 2-min maximal walking test; 10-m walking test; gait velocity at usual speed and fast speed; 4.44 m gait speed test; timed up and go test (TUG test); and 8-foot up and go test. Most of the studies reported improvements in the pre–post within-group and between-group (EG vs. CG). Nevertheless, Lee et al. [63] only found improvements in the pre–post in EG (4.44 m gait speed test and TUG test). Some studies [54,55,56] did not improve gait velocity at the usual speed. However, two of the three studies [54,55] presented improvements when comparing pre–post to gait velocity at fast speed. However, the significant between-group change in gait speed suggested that the clinical relevance is 0.05–0.10 m/s [103].

According to Ramírez-Campillo et al. [24], the HSRT intervention in older people significantly improved walking ability and TUG test results compared to the CG. Ringsberg et al. [104] and Karttunen et al. [105] reported that the older men and women (average 75 years old) showed a statistically significant relationship between isometric knee extensor strength and gait speed. Thereby, in this population, the HSRT with heavy and light loads seems to mediate adaptations in the neural system and translates into improvements in physical function, namely in gait function. In the same sense, Harridge et al. [106] reported submaximal voluntary activation of the knee extensor muscles in frail older people. The current knowledge indicates incomplete voluntary muscle activation as an essential mechanism for the loss of mechanical muscle function and physical function, contributing to the pathway of mobility disability [61,107,108].

In this meta-analysis, only two studies addressed the SPPB variable, which reduced the statistical power of these meta-analyses and did not allow an appropriate analysis of heterogeneity. They present high heterogeneity (*I*^2^ = 91%, 1.01 [−1.16, 3.18]); however, the variables show a prediction interval crossing the zero line. The same is observed in the gait velocity variable; although it was included in five studies, three of the studies had a major limitation, which was its origin in the same clinical trial [54,55,56], which could result in analysis bias. Finally, in the meta-analysis for the TUG test, only two studies reported this variable and presented values favorable to the intervention with the HSRT.

### 4.4. Body Composition

Finally, regarding body composition, the HSRT seems to induce skeletal muscle hypertrophy and neuromuscular adaptations [66]. This intervention can also facilitate increases in muscle quality in response to power training [59] and promote significant reductions in several parameters, namely body weight, body mass index, percent body fat, and waist-to-hip ratio [66]. Some studies showed evident morphological adaptations [58,109,110] due to an increase in muscle thickness of VL in all strength training groups, which indicates that hypertrophy likely contributed to the increased maximal and dynamic strength.

According to Reid et al. [59], the studies that reported improvements in muscle strength and power without associated alterations in body composition may reflect enhanced neural adaptations in the early stages of training, as reported by other studies [91,111]. It is likely that the earlier motor unit activation and enhanced maximal firing rates associated with HSRT would be principle stimuli for neural adaptations to this explosive form of training [111]. The results of improvements in muscle strength and power without associated alterations in body composition may also be a consequence of the short duration of interventions and low weekly frequency, and the differences in neuromuscular adaptation should be investigated over longer periods (e.g., six months).

### 4.5. Limitations

This study presents two limitations that should be pointed out. The first is the poor methodological quality of some of the included studies. Twelve of the fourteen included studies showed low methodological quality (level 2). This should be taken as a serious warning to the scientific community and may hinder the credibility and practical effect of said studies. Second, although our meta-analysis suggests very satisfactory results, it was conducted with a small number of studies in each function, so it is necessary that future studies be conducted in this area.

## 5. Conclusions

The present systematic review confirms that HSRT interventions that progressively increase in training intensity can improve several health outcomes in older people, mostly cognitive, neuromuscular, and physical function (moderate to large effects).

Furthermore, the results suggest that interventions with ten weeks or more and performed three times a week result in significant improvements in neuromuscular function.

## 6. Future Lines of Research

It should be noted that even though gait variability [97] and phase angle from bioelectrical impedance [112] have high clinical significance, none of the included studies verified the effect of the HSRT intervention on said outcomes. Therefore, we suggest that future studies conducted with older people and based on this approach (HSRT intervention) include the following variables. The first is gait variability—which refers to the magnitude of the stride-to-stride fluctuations and their changes over time during a walk—which may be useful in understanding the physiology of gait, in quantifying age-related and pathologic alterations in the locomotor control system, and in augmenting the objective measurement of mobility and functional status [113,114]. A gait variability study through non-linear methods [97,115] can help us to understand the physiological changes associated with ageing induced by an increase in noise in the neuromotor system, promoting significant changes in the control of the locomotor system [113,116]. Multiple age-related physiological changes increase neuromotor noise, increasing gait variability. If older adults alter how they regulate their stride variables, this could further exacerbate this variability. The second is the phase angle from electrical bioimpedance—the assessment of this indicator represents an interesting measure for future research on healthy populations, because it can reflect cellular health and mass and cell membrane integrity [112,117], and in the elderly population, it is a predictor of muscle function [96,118] and sarcopenia [119] and an indicator of general health status [120,121].

Finally, we also recommend further research to include an adequate follow-up (e.g., minimum six months) to assess the durability of HSRT intervention effects on all health-related variables.

## Figures and Tables

**Figure 1 ijerph-19-05390-f001:**
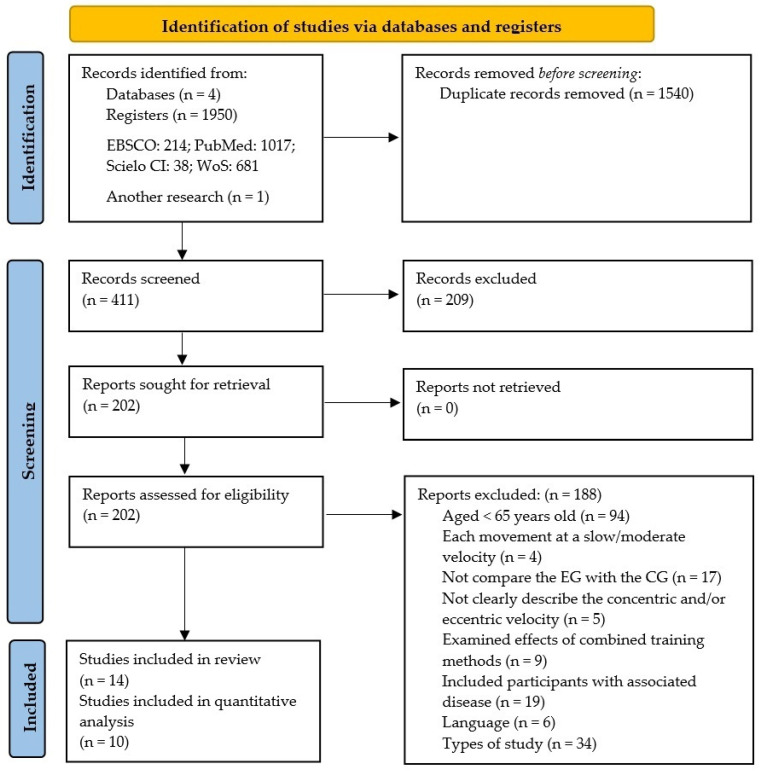
Flow chart of study design by PRISMA 2020.

**Figure 2 ijerph-19-05390-f002:**
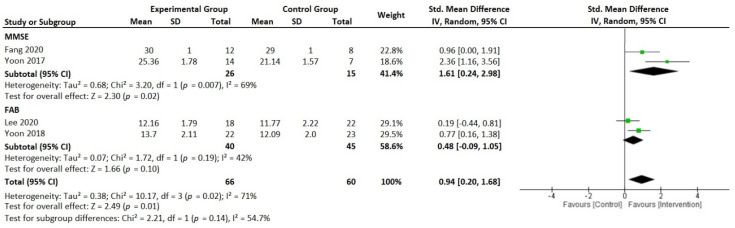
Forest plot presenting standardized mean difference and 95% confidence intervals from studies reporting high-speed resistance-training-induced changes in cognitive function between the intervention groups and the control groups. IV, independent variable; CI, confidence interval; SMD, standardized mean difference; FAB, frontal assessment battery; MMSE, mini-mental state examination [60,63,66,67].

**Figure 3 ijerph-19-05390-f003:**
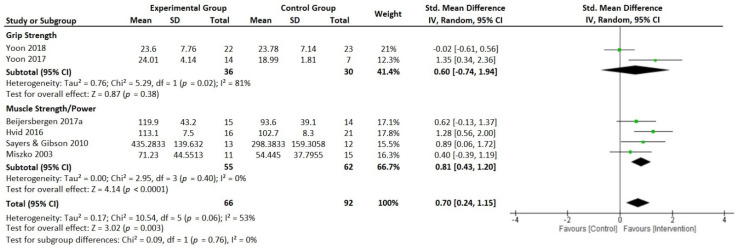
Forest plot presenting standardized mean difference and 95% confidence intervals from studies reporting high-speed resistance-training-induced changes in neuromuscular function between the intervention groups and the control groups. IV, independent variable; CI, confidence interval; SMD, standardized mean difference [54,57,61,65,66,67].

**Figure 4 ijerph-19-05390-f004:**
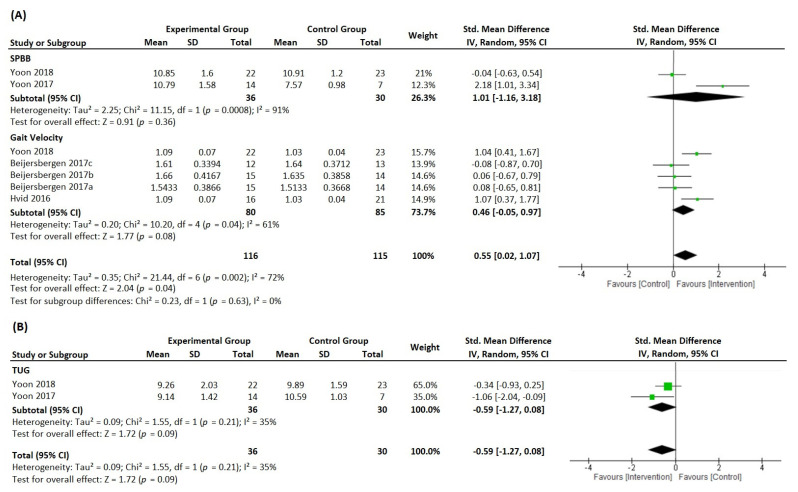
Forest plot presenting standardized mean difference and 95% confidence intervals from studies reporting high-speed resistance-training-induced changes in physical function ((**A**), through SPBB and gait velocity; (**B**), through TUG) between the intervention groups and the control groups. IV, independent variable; CI, confidence interval; SMD, standardized mean difference; SPPB, short physical performance battery; TUG, timed up and go test [54,55,56,61,66,67].

**Table 1 ijerph-19-05390-t001:** Analysis of the risk of bias of the studies included in this systematic review.

Authors	PEDro Scale											Total Score	Methodological Quality
	1	2	3	4	5	6	7	8	9	10	11		
Fang et al. [60] §	Y	1	0	0	0	0	0	1	0	1	1	4	**Moderate**
Lee et al. [63]	Y	1	0	1	0	0	0	1	0	1	1	5	**Moderate**
Yoon et al. [67]	N	1	0	1	0	0	0	0	0	1	1	4	**Moderate**
Yoon et al. [66]	Y	1	0	1	0	0	0	0	0	1	1	4	**Moderate**
Beijersbergen et al. [54] §	Y	0	0	1	0	0	0	1	0	0	1	3	**Poor**
Beijersbergen et al. [55] §	N	0	0	1	0	0	0	1	0	0	1	3	**Poor**
Beijersbergen et al. [56] §	N	0	0	1	0	0	0	1	0	0	1	3	**Poor**
Hvid et al. [61]	Y	1	0	1	0	0	1	0	0	1	1	5	**Moderate**
Sayers & Gibson [65]	N	1	0	1	0	0	0	0	0	1	1	4	**Moderate**
Marsh et al. [64]	Y	1	0	1	0	0	0	0	0	1	1	4	**Moderate**
Reid et al. [59]	Y	1	0	1	0	0	0	1	1	1	1	6	**Good**
Katula et al. [62]	Y	1	0	1	0	0	0	0	0	1	1	4	**Moderate**
Bean et al. [58]	Y	1	0	1	0	0	1	1	0	1	1	6	**Good**
Miszko et al. [57]	N	1	0	0	0	0	0	0	0	1	1	3	**Poor**

Abbreviations: 1, Eligibility; 2, Random allocation; 3, Concealed allocation; 4, Baseline comparability; 5, Blind subjects; 6, Blind therapists; 7, Blind assessors; 8, Adequate follow-up; 9, Intention-to-treat analysis; 10, Between-group comparisons; 11, Point estimates and variability; Y, yes; N, No; §, Scored by reviewers. Note: Eligibility criteria item does not contribute to total score.

**Table 2 ijerph-19-05390-t002:** Characteristics of included studies.

Authors	Country	Sample by Gender (N)	Age (years) (M ± SD)	Groups (N)	Body Fat (%)	Body Mass (kg)	BMI (kg/m^2^)	Registered Protocol?
Fang et al. [60]	USA	9 F 11 M	CG = 71.8 ± 4.8 EG = 70.8 ± 5.8	CG = 8 EG = 12	NR	NR	NR	No
Lee et al. [63]	South Korea	24 F 16 M	CG = 74.2 ± 4.4 EG = 73.7 ± 4.6	CG = 22 EG = 18	NR	CG = 60.6 EG = 61.0	NR	No
Yoon et al. [67]	South Korea	30 F 13 M	CG = 74.0 ± 4.3 EG = 73.9 ± 4.4	CG = 23 EG = 22	NR	NR	CG = 24.4 EG = 24.9	No
Yoon et al. [66]	South Korea	30 F	CG = 78.0 ± 1.0 EG = 75.0 ± 0.9	CG = 7 EG = 14	CG = 33.4 EG = 38.8	CG = 51.2 EG = 58.4	CG = 22.9 EG = 25.5	No
Beijersbergen et al. [54]	Germany	25 F 16 M	CG = 69.1 ± 4.4 EG = 72.9 ± 5.4	CG = 14 EG = 15	NR	CG = 73.9 EG = 73.6	CG = 25.5 EG = 25.8	No
Beijersbergen et al. [55]	Germany	29 *	CG = 69.1 ± 4.4 EG = 72.9 ± 5.4	CG = 14 EG = 15	NR	CG = 73.9 EG = 73.6	CG = 25.5 EG = 25.8	No
Beijersbergen et al. [56]	Germany	25 *	CG = 69.7 ± 5.0 EG = 72.1 ± 5.4	CG = 13 EG = 12	NR	NR	CG = 25.1 EG = 26.2	No
Hvid et al. [61]	Denmark	23 F 14 M	CG = 81.6 ± 1.1 EG = 82.3 ± 1.3	CG = 21 EG = 16	NR	CG = 73.4 EG = 76.5	NR	Yes
Sayers and Gibson [65]	USA	24 F 14 M	CG =72.8 ± 4.1 EG = 74.1 ± 6.4	CG = 12 EG = 13	NR	CG = 78.6 EG = 76.6	NR	No
Marsh et al. [64]	USA	25 F 11 M	CG = 74.4 ± 5.2 EG = 76.8 ± 6.4	CG = 15 EG = 15	NR	CG = 81.0 EG = 81.2	CG = 30.4 EG = 30.7	No
Reid et al. [59]	USA	31 F 26 M	CG = 79.7 ± 9.0 EG = 72.3 ± 6.0	CG = 12 EG = 23	NR	CG = 70.2 EG = 80.9	CG = 26.5 EG = 29.8	No
Katula et al. [62]	USA	36 *	CG = 74.3 ± 5.4 EG = 76.8 ± 6.5	CG = 13 EG = 12	NR	CG = 81.7 EG = 81.2	CG = 30.7 EG = 30.8	No
Bean et al. [58]	USA	21 F	CG = 78.9 ± 7.8 EG = 77.1 ± 5.7	CG = 10 EG = 11	NR	CG = 65.6 EG = 60.1	CG = 26.4 EG = 26.2	No
Miszko et al. [57]	Greece	22 F 17 M	CG = 72.4 ± 7.2 EG = 72.3 ± 6.7	CG = 15 EG = 11	CG = 26.8 EG = 29.1	CG = 68.2 EG = 79.7	NR	No

Abbreviations: N, number; M, mean; SD, standard deviation; CG, control group; EG, experimental group; F, female; M, male; USA, united states of America; BMI, body mass index; kg, kilograms; NR, not reported. *, date unreported gender data.

**Table 3 ijerph-19-05390-t003:** Description of the interventions performed in the included studies.

Authors	Exercise Modality	Exercises (Names)	Frequency (Days/ Week)	Intensity	Sets/Exercise (N)	Reps per Set (N)	Rest	Intervention Duration (weeks)	Session Duration (min)	Eccentric Velocity (s)	Supervised
Fang et al. [60]	Computerized Pneumatic Machines	Leg press; seated row; leg curl; chest press; hip adduction; lat pulldown; hip abduction; tricep extension; seated calf raise; bicep curl	3	40% 1 RM—lat pulldown 50% 1 RM—leg press, chest press, seated row 55% 1 RM—bicep curl 60% 1 RM—leg curl, tricep extension 65% 1 RM—calf raise, hip abduction, hip adduction	1 (wk 1) 2 (wk 2) 3 (wk 3–8)	10–12	1–2 min after each circuit	8	40–45	2	Yes
Lee et al. [63]	Elastic Bands	Seated row; one-leg press; applied pec deck flus or lateral raises; seated leg raises; squats; full squats; wide squats; bridging	3	12 to 13 from 6–20 RPE scale	2–3	10–12	30 s between exercises 1 min between sets	8	50	3	Yes
Yoon et al. [67]	Elastic Bands	Seated row; one leg press; applied pec deck flus; seated leg raises; lateral raise; semi-squats; wide squats; and bridging	3	12 to 13 from 6–20 RPE scale	2–3	12–15	NR	16	60	2	Yes
Yoon et al. [66]	Elastic Bands	NR	2	12 to13 from 6–20 RPE scale	2–3	12–15	2 min between exercises 1 min between sets	12	60	2	Yes
Beijersbergen et al. [54]	Weightlifting Machine	Leg press; ankle press; knee extension; knee flexion	3	40–60% 3 RM	3	6–10	NR	10	NR	Normal pace	NR
Beijersbergen et al. [55]	Weightlifting Machine	Leg press; ankle press; knee extension; knee flexion.	3	40–60% 3 RM	3	6–10	NR	10	NR	Normal pace	NR
Beijersbergen et al. [56]	Weightlifting Machine	Leg press; ankle press; knee extension; knee flexion	3	40–60% 3 RM	3	6–10	NR	10	NR	Normal pace	NR
Hvid et al. [61]	Weightlifting Machine	NR	2	70–80% 1 RM	3	10 (wk 1–6) 8 (wk 7–12)	NR	12	NR	2–3	Yes
Sayers and Gibson [65]	Computerized Pneumatic Machines	Leg press; seated knee extension	3	40% 1 RM	3	12–14	NR	12	NR	2	Yes
Marsh et al. [64]	Computerized Pneumatic Machines	Leg press; knee extension	3	70% 1 RM	3	8–10	NR	12	60	2–3	Yes
Reid et al. [59]	Computerized Pneumatic Machines	Leg press; knee extension	3	70% 1 RM	3	8	NR	12	NR	>2	Yes
Katula et al. [62]	Computerized Pneumatic Machines	Leg press; knee extension	3	70% 1 RM	3	8–10	NR	12	NR	2–3	Yes
Bean et al. [58]	Weighted Vest	Chair stands; toe raises; pelvic raises; step ups; seated tricep dips; chest press	3	At 16 from 6–20 RPE scale	3	10	1–2 min between sets	12	30	>2	Yes
Miszko et al. [57]	Computerized Pneumatic Machines	Seated row; chest press; tricep extension; leg press; leg extension; seated leg curl; bicep curls; plantar flexion; jump squats	3	40–70% 1 RM	3	8–10	NR	16	NR	2	NR

Abbreviations: N, number; min, minutes; s, seconds; wk, week; RM, repetition maximum; RPE, rated perceived exertion; NR, not reported.

**Table 4 ijerph-19-05390-t004:** Main results of the included studies.

Authors	Aim	Outcome Variable	Group	Main Results (M ± SD)		*p* Value	*p* Value (CG vs. EG) ^&^	Effect Size ^a^
				Pre	Post			
Fang et al. [60]	To characterize changes in the retinal microvascular density and their relationship with cognitive function in cognitively normal older people	Outcomes of optical function						
		Vessel density in the total retinal vascular network	CG	1.79	1.79	>0.05	>0.05	NR
			EG	1.79	1.79	>0.05		NR
		Vessel density in the superficial vessel plexus	CG	1.79	1.78	>0.05	>0.05	NR
			EG	1.78	1.78	>0.05		NR
		Vessel density in the deep vessel plexus	CG	1.80	1.80	>0.05	>0.05	NR
			EG	1.80	1.80	>0.05		NR
		Outcome of cognitive function						
		MMSE (score)	CG	30 ± 0	29 ± 1	**0.08**	NR	NR
			EG	30 ± 1	30 ± 1	0.59		NR
Lee et al. [63]	To investigate the effects of HSRT on neuromuscular, executive, and gait performance	Outcomes of neuromuscular function						
		Movement time (ms)	CG	906.36 ± 36.27	1010.04 ± 65.89	>0.05	**<0.05**	NR
			EG	921.69 ± 40.10	799.51 ± 72.84	**<0.05**		NR
		Pre-motor time (ms)	CG	700.64 ± 36.62	757.34 ± 56.37	>0.05	>0.05	NR
			EG	663.82 ± 40.48	626.96 ± 62.32	>0.05		NR
		Motor time (ms)	CG	205.67 ± 17.45	252.70 ± 34.44	>0.05	**<0.05**	NR
			EG	271.40 ± 19.29	181.15 ± 38.08	**<0.05**		NR
		Antagonist co-activation (% *p*. EMG)	CG	25.28 ± 3.75	25.04 ± 3.72	>0.05	>0.05	NR
			EG	27.89 ± 4.14	26.32 ± 3.48	>0.05		NR
		Rate of EMG rise (% *p*. EMG/s^−1^)	CG	169.30 ± 12.04	161.82 ± 10.41	>0.05	**<0.05**	NR
			EG	166.48 ± 13.31	197.94 ± 11.51	**<0.05**		NR
		Normalized peak torque (N·m·kg^−1^)	CG	NR	NR	>0.05	>0.05	NR
			EG	NR	NR	**<0.05**		NR
		Rate of torque development	CG	NR	NR	>0.05	**<0.05**	NR
			EG	NR	NR	**<0.01**		NR
		Outcome of cognitive function						
		FAB (score)	CG	11.72 ± 2.11	11.77 ± 2.22	>0.05	>0.05	NR
			EG	11.38 ± 2.56	12.16 ± 1.79	>0.05		NR
		Outcomes of physical function						
		4.44 m gait speed (s)	CG	NR	NR	>0.05	>0.05	NR
			EG	NR	NR	**<0.01**		NR
		TUG (s)	CG	NR	NR	>0.05	>0.05	NR
			EG	NR	NR	**<0.01**		NR
Yoon et al. [67]	To elucidate the effects of high-speed resistance exercise on cognitive function and physical performance	Outcomes of cognitive function						
		Memory (score)	CG	10.26 ± 2.85	10.52 ± 2.79	>0.05	0.445	0.09
			EG	8.55 ± 2.39	10.00 ± 3.71	>0.05		0.46 #
		Processing speed (s)	CG	43.04 ± 11.95	42.59 ± 15.92	>0.05	**0.036**	0.03
			EG	54.15 ± 28.43	48.26 ± 27.33	**<0.05**		0.21 #
		Cognitive flexibility (s)	CG	188.92 ± 81.38	187.20 ± 70.14	>0.05	0.532	0.45 #
			EG	163.37 ± 62.45	140.82 ± 34.65	>0.05		0.02
		Working memory (score)	CG	10.09 ± 2.04	10.39 ± 1.83	>0.05	0.448	0.14
			EG	10.20 ± 1.54	10.70 ± 1.34	>0.05		0.35 #
		FAB (score)	CG	11.87 ± 2.12	12.09 ± 2.00	>0.05	**0.022**	0.11
			EG	12.00 ± 2.45	13.70 ± 2.11	**<0.05**		0.74 *
		Outcomes of physical function						
		SPBB (score)	CG	10.04 ± 1.46	10.91 ± 1.20	>0.05	**0.001**	0.65 *
			EG	9.25 ± 2.31	10.85 ± 1.60	**<0.05**		0.81 *
		TUG (s)	CG	9.95 ± 1.51	9.89 ± 1.59	>0.05	**<0.01**	0.04
			EG	10.66 ± 2.41	9.26 ± 2.03	**<0.01**		0.65 *
		4.44 m gait speed (s)	CG	6.04 ± 0.82	5.58 ± 0.81	>0.05	**0.027**	0.56 #
			EG	6.21 ± 1.04	5.34 ± 0.81	**<0.01**		0.93 *
		Outcomes of neuromuscular function						
		Grip strength (kg)	CG	21.81 ± 6.31	23.78 ± 7.14	>0.05	**0.020**	0.29 #
			EG	21.41 ± 6.58	23.60 ± 7.76	**<0.05**		0.30 #
		Isokinetic 60°/s peak torque/BW	CG	70.77 ± 24.32	64.23 ± 20.72	>0.05	**0.004**	0.01
			EG	65.05 ± 25.82	71.20 ± 36.68	**<0.05**		0.19
		Isokinetic 180°/s average power per rep (W)	CG	72.77 ± 23.82	66.59 ± 23.67	>0.05	**0.001**	0.26 #
			EG	68.32 ± 40.60	82.09 ± 44.63	**<0.05**		0.32 #
Yoon et al. [66]	To compare the effects of two different exercise regimens on cognitive function, body composition, muscular strength, and functional ability	Outcomes of body composition						
		BW (kg)	CG	51.19 ± 4.13	50.14 ± 3.75	**<0.01**	**0.001**	NR
			EG	58.39 ± 6.82	57.50 ± 7.01	**<0.01**		NR
		BMI(kg/m^2^)	CG	22.90 ± 1.81	22.54 ± 1.71	>0.05	**0.005**	NR
			EG	25.46 ± 2.47	25.02 ± 2.35	**<0.01**		NR
		Skeletal muscle mass (kg)	CG	17.11 ± 1.45	17.49 ± 1.35	>0.05	**0.008**	NR
			EG	18.84 ± 2.84	19.63 ± 2.63	**<0.01**		NR
		Percent body fat (%)	CG	33.39 ± 8.26	33.27 ± 3.35	>0.05	**0.004**	NR
			EG	38.83 ± 5.37	35.50 ± 4.72	**<0.01**		NR
		WHR	CG	0.88 ± 0.04	0.86 ± 0.03	>0.05	**0.014**	NR
			EG	0.91 ± 0.07	0.89 ± 0.06	**<0.05**		NR
		Arm circumference (cm)	CG	26.71 ± 2.75	26.29 ± 2.69	>0.05	0.494	NR
			EG	29.21 ± 2.91	28.04 ± 2.32	>0.05		NR
		Thigh circumference (cm)	CG	49.14 ± 4.60	47.57 ± 4.71	>0.05	0.115	NR
			EG	51.14 ± 4.00	50.57 ± 4.69	>0.05		NR
		Outcomes of cognitive function						
		MMSE (score)	CG	22.29 ± 1.11	21.14 ± 1.57	**<0.05**	**<0.05**	−0.85
			EG	21.00 ± 1.04	25.36 ± 1.78	**<0.01**		2.99 §
		MoCA (score)	CG	18.71 ± 2.63	18.14 ± 2.97	>0.05	**<0.05**	0.19
			EG	18.29 ± 2.81	24.29 ± 2.58	**<0.01**		2.22 §
		Outcomes of physical function						
		SPPB (score)	CG	7.14 ± 1.77	7.57 ± 0.98	>0.05	**<0.05**	0.30 #
			EG	8.14 ± 2.48	10.79 ± 1.58	**<0.01**		1.27 §
		TUG (s)	CG	11.48 ± 1.02	10.59 ± 1.03	>0.05	>0.05	−0.87
			EG	0.51 ± 1.86	9.14 ± 1.42	>0.05		−0.83
		Outcomes of neuromuscular function						
		Grip strength (kg)	CG	17.69 ± 0.91	18.99 ± 1.81	>0.05	**<0.05**	1.86 §
			EG	19.26 ± 3.57	24.01 ± 4.14	**<0.01**		4.58 £
		Isokinetic 60°/s peak torque/BW—Right extensor (Nm)	CG	105.71 ± 20.92	116.86 ± 31.74	>0.05	>0.05	0.41 #
			EG	77.57 ± 29.03	115.43 ± 31.92	**<0.01**		1.24 §
		Isokinetic 60°/s peak torque/BW—Right flexor (Nm)	CG	74.86 ± 17.44	58.14 ± 18.00	**<0.05**	>0.05	−0.94
			EG	66.21 ± 20.77	67.43 ± 17.82	>0.05		0.06
		Isokinetic 60°/s peak torque/BW—left extensor (Nm)	CG	96.86 ± 25.42	108.86 ± 27.67	>0.05	>0.05	0.45 #
			EG	90.36 ± 31.70	90.57 ± 21.99	>0.05		0.01
		Isokinetic 60°/s peak torque/BW—left flexor (Nm)	CG	69.57 ± 19.84	53.00 ± 16.34	**<0.05**	>0.05	−0.91
			EG	64.14 ± 17.22	58.10 ± 17.60	>0.05		−0.35
		Isokinetic 180°/s peak torque/BW—Right extensor (Nm)	CG	57.29 ± 13.88	67.43 ± 18.60	**<0.05**	**<0.05**	0.62 *
			EG	49.07 ± 20.00	68.29 ± 17.55	**<0.01**		1.02 *
		Isokinetic 180°/s peak torque/BW—Right flexor (Nm)	CG	54.43 ± 9.24	37.71 ± 9.93	>0.05	>0.05	−1.74
			EG	44.14 ± 14.69	36.00 ± 15.11	>0.05		−0.55
		Isokinetic 180°/s peak torque/BW—Left extensor (Nm)	CG	54.71 ± 14.85	66.57 ± 21.45	**<0.05**	>0.05	0.64 *
			EG	50.29 ± 17.82	63.21 ± 19.88	**<0.05**		0.68 *
		Isokinetic 180°/s peak torque/BW—Left flexor (Nm)	CG	51.43 ± 13.13	32.57 ± 13.01	**<0.01**	>0.05	−1.44
			EG	45.71 ± 14.31	37.22 ± 13.79	**<0.05**		−0.60
Beijersbergen et al. [54]	To determine the effects of lower-extremity power training and detraining on lower-limb muscle power and gait kinematics	Outcomes of neuromuscular function						
		Maximal muscle power of knee extension 60°/s (W)	CG	98.4 ± 39.4	93.6 ± 39.1	0.680	NR	−0.12
			EG	97.5 ± 37.7	119.9 ± 43.2	**<0.01**		0.59 #
		Maximal muscle power of knee extension 120°/s (W)	CG	169.4 ± 61.7	164.7 ± 49.2	0.738	NR	−0.08
			EG	161.7 ± 64.5	199.1 ± 72.8	**<0.01**		0.58 #
		Maximal muscle power of knee extension 180°/s (W)	CG	229.2 ± 77.6	234.0 ± 64.9	0.685	NR	0.06
			EG	216.9 ± 93.5	256.9 ± 96.3	**<0.01**		0.43 #
		Maximal muscle power of knee flexion 60°/s (W)	CG	54.6 ± 24.6	51.1 ± 20.4	0.557	NR	−0.14
			EG	54.6 ± 27.7	71.5 ± 37.5	**<0.01**		0.61 *
		Maximal muscle power of knee flexion 120°/s (W)	CG	103.4 ± 38.9	101.0 ± 35.5	0.794	NR	−0.06
			EG	104.6 ± 59.0	126.8 ± 71.1	**<0.01**		0.38 #
		Maximal muscle power of knee flexion 180°/s (W)	CG	156.6 ± 59.0	161.7 ± 50.3	0.541	NR	0.09
			EG	167.7 ± 92.2	186.3 ± 96.9	**0.002**		0.20 #
		Maximal muscle power of knee plantar flexion 20°/s (W)	CG	13.6 ± 6.7	15.3 ± 6.9	0.095	NR	0.25
			EG	12.6 ± 8.1	17.3 ± 9.8	**0.001**		0.57 #
		Maximal muscle power of knee plantar flexion 40°/s (W)	CG	25.3 ± 11.2	29.1 ± 14.7	0.225	NR	0.34 #
			EG	23.3 ± 13.6	30.1 ± 16.0	**<0.01**		0.50 #
		Maximal muscle power of knee plantar flexion 60°/s (W)	CG	35.8 ± 15.8	36.6 ± 19.6	0.734	NR	0.05
			EG	32.4 ± 21.1	40.8 ± 23.2	**<0.01**		0.40 #
		Outcomes of physical function						
		Stair ascent power (W·kg^−1^)	CG	4.36 ± 0.65	4.41 ± 0.85	0.819	NR	0.07
			EG	4.05 ± 0.84	4.36 ± 0.92	0.075		0.38 #
		Stair descent power (W·kg^−1^)	CG	5.01 ± 0.91	5.21 ± 1.18	0.225	NR	0.22 #
			EG	4.48 ± 0.87	4.88 ± 1.21	0.061		0.46 #
		Six-min walk test (m/s)	CG	1.26 ± 0.14	1.27 ± 0.14	0.293	NR	0.03
			EG	1.29 ± 0.14	1.31 ± 0.15	0.252		0.18
		Gait velocity (habitual speed) (m/s)	CG	1.35 ± 0.14	1.34 ± 0.16	0.652	NR	−0.07
			EG	1.32 ± 0.16	1.36 ± 0.15	0.220		0.25 #
		Gait velocity (fast speed) (m/s)	CG	1.97 ± 0.35	1.93 ± 0.31	0.526	NR	−0.18
			EG	1.85 ± 0.28	1.96 ± 0.38	**0.026**		0.39 #
Beijersbergen et al. [55]	To examine the effects of lower-extremity power training and detraining on gait kinetics	Outcomes of neuromuscular function						
		Knee extensor power 60 deg/s (W/kg·m)	CG	NR	NR	>0.05	NR	NR
			EG	NR	NR	**<0.05**		NR
		Knee extensor power 120 deg/s (W/kg·m)	CG	NR	NR	>0.05	NR	NR
			EG	NR	NR	**<0.05**		NR
		Knee extensor power 180 deg/s (W/kg·m)	CG	NR	NR	>0.05	NR	NR
			EG	NR	NR	**<0.05**		NR
		Plantar flexor power 20 deg/s (W/kg·m)	CG	NR	NR	>0.05	NR	NR
			EG	NR	NR	**<0.05**		NR
		Plantar flexor power 40 deg/s (W/kg·m)	CG	NR	NR	>0.05	NR	NR
			EG	NR	NR	**<0.05**		NR
		Plantar flexor power 60 deg/s (W/kg·m)	CG	NR	NR	>0.05	NR	NR
			EG	NR	NR	**<0.05**		NR
		Outcomes of physical function						
		Gait velocity (habitual speed) (m/s)	CG	1.35 ± 0.14	1.34 ± 0.16	0.652	NR	NR
			EG	1.32 ± 0.16	1.36 ± 0.15	0.220		NR
		Gait velocity (fast speed) (m/s)	CG	1.97 ± 0.35	1.93 ± 0.31	0.526	NR	NR
			EG	1.85 ± 0.28	1.96 ± 0.38	**0.026**		NR
Beijersbergen et al. [56]	To examine the effects of 10 weeks of lower-extremity power training on gait velocity and neuromuscular activation of lower-extremity muscles during level walking	Outcomes of neuromuscular function						
		Isometric muscle strength of the knee flexors	CG	NR	NR	>0.05	NR	≤0.31
			EG	NR	NR	**0.002**		0.34 #
		Isometric muscle strength of the knee extensors	CG	NR	NR	**0.021**	NR	0.50 #
			EG	NR	NR	**<0.01**		0.74 *
		Isometric muscle strength of the plantar flexors	CG	NR	NR	>0.05	NR	NR
			EG	NR	NR	**0.002**		0.80 *
		EMG amplitudes of the knee flexors	CG	NR	NR	>0.05	NR	NR
			EG	NR	NR	**0.004**		1.47 §
		EMG amplitudes of the knee extensors	CG	NR	NR	**0.021**	NR	NR
			EG	NR	NR	**0.013**		0.50 #
		EMG amplitudes of the plantar flexors	CG	NR	NR	>0.05	NR	NR
			EG	NR	NR	0.076		0.47 #
		Outcomes of physical function						
		Gait velocity (habitual speed) (m/s)	CG	1.41 ± 0.19	1.39 ± 0.29	0.240	NR	−0.11
			EG	1.28 ± 0.14	1.34 ± 0.17	0.079		0.42 #
		Gait velocity (fast speed) (m/s)	CG	1.91 ± 0.26	1.89 ± 0.26	0.396	NR	−0.07
			EG	1.82 ± 0.18	1.88 ± 0.23	0.059		0.31 #
Hvid et al. [61]	To examine the effects of 12 weeks of progressive high-intensity power training on the outcomes of knee extensor voluntary muscle activation and maximal gait speed	Outcomes of neuromuscular function						
		Thickness (cm)	CG	2.52 ± 0.11	2.50 ± 0.11	>0.05	>0.05	NR
			EG	2.69 ± 0.12	2.74 ± 0.15	>0.05		NR
		Strength (N. m)	CG	101.9 ± 0.1	102.7 ± 8.3	>0.05	**<0.05**	NR
			EG	98.9 ± 7.7	113.1 ± 7.5	**<0.05**		NR
		Voluntary activation (%)	CG	79.1 ± 2.4	79.0 ± 2.6	>0.05	**<0.05**	NR
			EG	78.9 ± 3.5	84.9 ± 2.1	**<0.05**		NR
		Outcomes of physical function						
		2-MWT (m/s)	CG	1.06 ± 0.04	1.03 ± 0.04	**<0.05**	**<0.05**	NR
			EG	1.00 ± 0.06	1.09 ± 0.07	**<0.05**		NR
Sayers and Gibson [65]	To determine whether low-load HSPT and traditional high-load, slow-speed resistance training may have differing effects on power output obtained across a range of external resistances and to explore the impact of these RT regimens on the determinants of muscle power	Outcomes of neuromuscular function						
		Peak power 40% 1 RM (W)	CG	292.6 ± 165.9	283.5 ± 153.9	NR	**<0.05**	NR
			EG	313.2 ± 118.4	383.3 ± 113.3	NR		NR
		Peak power 50% 1 RM (W)	CG	315.5 ± 180.0	313.9 ± 169.1	NR	**≤0.02**	NR
			EG	339.0 ± 138.1	426.6 ± 133.6	NR		NR
		Peak power 60% 1 RM (W)	CG	315.4 ± 178.8	327.3 ± 180.0	NR	**≤0.03**	NR
			EG	350.6 ± 144.1	456.6 ± 136.6	NR		NR
		Peak power 70% 1 RM (W)	CG	304.4 ± 174.3	320.3 ± 180.5	NR	**<0.01**	NR
			EG	342.8 ± 149.6	456.3 ± 150.2	NR		NR
		Peak power 80% 1 RM (W)	CG	281.4 ± 155.3	302.9 ± 169.7	NR	**≤0.02**	NR
			EG	317.2 ± 145.6	457.6 ± 149.3	NR		NR
		Peak power 90% 1 RM (W)	CG	252.1 ± 150.7	242.4 ± 111.2	NR	**≤0.04**	NR
			EG	272.9 ± 154.2	431.3 ± 162.4	NR		NR
		Velocity at peak power (N·m·s^−1^)	CG	NR	NR	NR	**<0.01**	NR
			EG	NR	NR	NR		NR
Marsh et al. [64]	To compare the effects of lower-extremity power training on muscle strength, physical function, and body composition	Outcomes of neuromuscular function						
		Knee extension strength (kg)	CG	25.03 ± 10.03	23.99 ± NR	NR	**0.024**	NR
			EG	25.87 ± 11.77	29.33 ± NR	NR		NR
		Knee extension power (W)	CG	161.46 ± 62.44	143.60 ± NR	NR	**0.003**	NR
			EG	148.33 ± 84.50	221.30 ± NR	NR		NR
		Leg press strength (kg)	CG	93.97 ± 31.17	100.12 ± NR	NR	**0.026**	NR
			EG	94.07 ± 34.09	117.82 ± NR	NR		NR
		Leg press power (W)	CG	185.47 ± 75.64	211.4 ± NR	NR	**<0.01**	NR
			EG	171.06 ± 89.04	308.7 ± NR	NR		NR
Reid et al. [59]	To explore the effects of power training on muscle power and strength	Outcomes of neuromuscular function						
		Knee extension 1 RM—∆ (W)	CG	NR	NR	NR	**<0.01**	NR
			EG	NR	NR	**<0.01**		NR
		Knee extension—absolute peak power at 40% 1 RM (W)	CG	NR	NR	NR	**≤0.003**	NR
			EG	48 ± 15	62 ± 16	**0.002**		NR
		Knee extension—absolute peak power at 70% 1 RM (W)	CG	NR	NR	NR	**≤0.003**	NR
			EG	77 ± 39	118 ± 54	**<0.05**		NR
		Leg press 1 RM—∆ (W)	CG	NR	NR	NR	>0.05	NR
			EG	NR	NR	0.140		NR
		Leg press—absolute peak power at 40% 1 RM (W)	CG	NR	NR	NR	>0.05	NR
			EG	NR	NR	0.190		NR
		Leg press peak power at 70% 1 RM—∆ (W)	CG	NR	NR	NR	>0.05	NR
			EG	NR	NR	0.220		NR
Katula et al. [62]	To compare the effects of strength training and power training to one another and to a wait list control group with respect to changes in quality of life	Outcomes of quality of life						
		Self-efficacy for strength (score)	CG	25.07 ± 17.67	34.46 ± 19.88	>0.05	**<0.05**	0.50 #
			EG	25.07 ± 17.53	72.16 ± 22.58	**<0.05**		2.34 £
		Satisfaction with physical function (score)	CG	−0.73 ± 1.69	−0.35 ± 1.96	>0.05	**<0.05**	0.21 #
			EG	−0.85 ± 1.86	1.10 ± 1.20	**<0.05**		1.21 §
		Life satisfaction (score)	CG	22.93 ± 6.46	21.46 ± 6.06	>0.05	**<0.05**	−0.23
			EG	25.78 ± 7.47	29.25 ± 6.38	**<0.05**		0.50 #
Bean et al. [58]	To evaluate the efficacy of another form of weighted vest exercise on muscle power and mobility function	Outcomes of physical function						
		SPBB (score)	CG	7.30 ± 1.50	NR	**0.009**	0.377	NR
			EG	7.70 ± 1.30	NR	**<0.01**		NR
		Chair-5 time (s)	CG	19.60 ± 4.10	NR	**<0.01**	**0.019**	NR
			EG	18.50 ± 3.60	NR	**<0.01**		NR
		Gait speed (m/s)	CG	0.70 ± 0.16	NR	0.339	0.356	NR
			EG	0.80 ± 0.15	NR	**0.006**		NR
		Unilateral stance time (s)	CG	6.05 ± 5.90	NR	0.900	0.342	NR
			EG	4.52 ± 5.40	NR	**0.028**		NR
Miszko et al. [57]	To determine whether power training was more efficacious than strength training for improving whole-body physical function and to examine the relationship between changes in anaerobic power and muscle strength and changes in physical function	Outcomes of neuromuscular function						
		Chest press—1 RM (kg)	CG	29.36 ± 12.20	29.18 ± 13.60	NR	>0.05	NR
			EG	31.01 ± 12.90	34.81 ± 14.60	NR		NR
		Leg press—1 RM (kg)	CG	75.61 ± 38.90	79.71 ± 37.50	NR	>0.05	NR
			EG	95.45 ± 33.20	107.65 ± 32.20	NR		NR
		Outcomes of physical function						
		CS-PFP test—total (score)	CG	55.5 ± 14	57.0 ± 18	NR	**<0.05**	NR
			EG	58.2 ± 13	67.1 ± 13	NR		NR

Abbreviations: Significant differences are highlighted in bold. CG, control group; EG, experimental group; M, mean; SD; standard deviation; NR, not reported; MMSE, mini-mental state test; FAB, frontal assessment battery; SPPB, short physical performance battery; TUG, time up and go test; ms, milliseconds; s, seconds; kg, kilograms; BW, body weight; Nm, newton meter; m/s, meter per second; deg/s, degree per second; mm, millimeter; cm, centimeters; BMI, body mass index; WHR, waist-to-hip ratio; MoCA, cognitive assessment; W, watts; EMG, surface electromyography for maximal muscle activation; RFD, rate of force development; CS-PFP, continuous-scale physical functional performance; N, newton; RM, repetition maximum; HSRT, high-speed resistance training; μV, microvolt; rep, repetitions; ^&^, *p* value of comparison between CG and EG at post; ^a^, effect size by Hopkins et al. [68]; #, small effect size (between 0.20 and 0.59); *, moderate effect size (between 0.60 and 1.19); §, large effect size (between 1.20 and 1.99); £, very large effect size (≥2.00).

## Data Availability

Not applicable.

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
