# Peer review of "The Effects of High-Speed Resistance Training on Health Outcomes in Independent Older Adults: A Systematic Review and Meta-Analysis"

_ijerph, 2022, doi:10.3390/ijerph19095390_

Round 1

Reviewer 1 Report

Thank you for the opportunity to review your manuscript,   The effects of high-speed resistance training on health out comes in independent older adults: a systematic review

The purpose of this study was to aimed to systematically review randomized controlled trials that investigated the general impacts of a high-speed resistance training (HSRT) protocols on health outcomes within an ageing population.

The abstract needs to be reworded. It focuses on the study selection process without briefly outlining the problems that generated this study.

Exclusion criteria should not be non-compliance with exclusion criteria.

The conclusions should focus on the findings and not on the interpretation of the findings. If the authors feel that their interpretation is essential, it should be included in the discussion.

Lines 455-456, I think, should be deleted.

Author Response

Dear Reviewer,

I am enclosing the responses to your comments.

Reviewer 2 Report

I really enjoyed reviewing this paper. I believe that this is an important topic for many health and sports science professionals. This manuscript presents a high rigor. It is a pity that most of the studies analyzed present a moderate to poor methodological quality. I would recommend the authors to add a small meta-analysis on some of the variables discussed in discussion. It would give a greater manuscript relevance.
Introduction:
I think this section is very accurate. Since it presents relevant information in a simple and brief way, generating a good theoretical framework for the reader.
Material and methods
In this section the authors comment that the review is from the most recent literature. Has any exclusion criteria been used based on the date of publication? Otherwise, the methodology section is very well designed.
Have you excluded all articles that were not in English? Why haven't you included articles in Portuguese or Spanish?
Results
In table 2, I would keep the format (23 F 24M), I would not put 23 F and M.
Discussion
Too long, I would try to summarize it a bit.

Author Response

Dear Reviewer

I am enclosing the responses to your comments.

Reviewer 3 Report

I congratulate the authors for the systematic review. It seems to me a solid work, well written and updated methodology. 

I would only have one recommendation, wich would be to delve into adherence to physical activity programs, wich I could add to the discussion (since they presented something in the results), due to the importance of said variable in the programs oriented to the older people.

Author Response

(The authors gave the same response as above.)

Round 2

Reviewer 2 Report

The authors made the established modifications. I still believe that a systematic review with meta-analysis could be carried out on some of the variables analyzed.

Author Response

Dear Reviewer,

Thank you very much for your suggestions.
I am enclosing the response to your comment.
